# A Qualitative Exploration of Stakeholder Involvement in Decision-Making for Alcohol Treatment and Prevention Services

**DOI:** 10.3390/ijerph19042148

**Published:** 2022-02-14

**Authors:** Hayley Alderson, Eileen Kaner, Amy O’Donnell, Angela Bate

**Affiliations:** 1Population Health Sciences Institute, Newcastle University, Newcastle upon Tyne NE2 4AX, UK; eileen.kaner@newcastle.ac.uk (E.K.); amy.odonnell@newcastle.ac.uk (A.O.); 2Coach Lane Campus West, Northumbria University, Newcastle upon Tyne NE7 7XA, UK; angela.bate@northumbria.ac.uk

**Keywords:** stakeholder involvement, decision making, alcohol

## Abstract

The concept of providing individuals with a ‘voice’ via stakeholder involvement has been advocated within English health care policy for several decades. Stakeholder involvement encourages people affected by an issue to contribute to planning and decision making regarding treatment and care, inclusive of providers and recipients of care. This paper explores stakeholder involvement in the design and delivery of public health alcohol services. A qualitative case study approach was adopted, including in-depth interviews with 11 alcohol commissioners, 10 alcohol service providers and 6 general practitioners plus three facilitated focus groups with 31 alcohol service users. Findings show that most participants were aware of, and could name, various methods of stakeholder involvement that they had engaged with; however, the extent and impact of stakeholder involvement in decision making are not transparent. It is essential that a deeper understanding is generated of the different roles that stakeholders can play within the entire decision-making process to maximise its utility.

## 1. Introduction

In the U.K., there were 270,705 adults in contact with drug and alcohol treatment services between April 2019 and March 2020 [1]. Of the population accessing treatment, 140,599 were receiving treatment for opiate use (the largest group) and 74,618 were receiving treatment for alcohol use (the second largest group). When considering treatment interventions delivered, 98% of individuals received a psychosocial intervention, and 56% received at least one pharmacological intervention [1]. Within the same 2019–2020 timeframe, 117,678 individuals exited the treatment system, of which 36% left without completing treatment (this includes individuals declining further treatment and unsuccessful transfers between services) [1]. The significant number of people not successfully completing their treatment as planned can result in high costs for the individual, their family, and the community [2]. Engaging in effective stakeholder involvement to collaboratively bring about improvements in alcohol service design and delivery and ensuring a clear understanding of the factors that promote or inhibit drug and alcohol treatment uptake and attendance at social and psychological interventions are crucial. Involving stakeholders in service design offers is one way of improving service provision, including determining which interventions are perceived to contribute to reducing the alcohol- and drug-related harm from individuals’ experience.

A stakeholder can be defined as a group of people or an individual who has a ‘stake’ in a situation. This can include individuals who participate in the co-ordination, resourcing and delivery of services such as commissioners and collaborators and individuals such as service users and members of the community who contribute their views and experiences of services to influence service design [3]. The routine inclusion of stakeholders (including patient and public involvement) in the organisation, design and delivery of health services is considered good practice. In recent years, the role of public and patient involvement regarding improvement in health care quality [4], patient safety [5] health service development [6], intervention development for mental health conditions such as depression and paranoia [7,8] and within research studies [9,10] has been discussed.

In an attempt to become more responsive to the needs of the population, the concept of providing individuals with a ‘voice’ via stakeholder involvement has been advocated within English health care policy for several decades [11,12,13]. Moves to increase stakeholder involvement in decision making have accelerated in recent years in response to two major factors, one being that politicians demand greater efficiency from services and an effective use of public funds and, secondly, due to members of the public demanding increased involvement in decisions about their care and services [14,15].

The ‘Transforming Participation in Health and Care’ [12], the NHS England guidance ‘Patient and Public participation: in commissioning health and care’, and the 2019 NHS Long Term Plan [16] have set out key actions on how to embed the involvement of members of the public and stakeholders in their work [13]. Indeed, the ‘Think Local act Personal’ partnership shows the support which exists for greater personalisation of health and social care [17] where stakeholder involvement inclusive of individual service users have a greater role in policy making and shared decision making in health and care settings [18].

There are several perceived benefits of involving stakeholders in decision-making processes. Stakeholder involvement has the potential to present everyone involved in decision making to different perceptions and standpoints which facilitate a broader understanding to be obtained regarding the community context that exists [19,20,21]. Additionally, it may strengthen democracy and enhance the credibility of services by promoting an understanding of the issues, reducing uncertainty, and promoting trust and legitimacy on behalf of service users [22,23] which in turn could improve the sustainability of public health services if stakeholders have confidence that services are endeavouring to meet their needs [24,25,26]. Elliott and Williams [27] have argued that the experiential wisdom that lay people hold is a form of legitimate expertise which can become ‘the basis for a powerful form of knowledge production’ adding an extra dimension within the decision-making process [22]. A lay person is a person who is not trained, qualified, or experienced in a particular subject or activity [28].

Despite the perceived benefits of stakeholder involvement, bringing together multiple stakeholders is not without problems [29]. The process of involving stakeholders in decision-making processes can be complicated as the identity of individual stakeholders is not fixed, the level of interest an individual has regarding an issue and the ability to influence may fluctuate. The multifaceted nature of alcohol use and misuse lends itself to individuals encompassing multiple identities where needs are managed across statutory and voluntary services covering both health and social needs. This highlights a challenge for decision makers and treatment services alike when considering which intervention options to offer to meet service user needs across the entire prevention, treatment, and recovery continuum.

In addition, it is well recognised that hierarchies of control and influence exist within and between stakeholders [30,31]. Constructive relationships are required to enable meaningful dialogue to be undertaken; however, the decision-making process often involves bringing together stakeholders with multiple, often conflicting, views or priorities, pursuing different agendas [32]. Stakeholders occupy different levels of knowledge and expertise and the views of stakeholders are not weighted equally. It is stated by Foucault [33] that institutional hierarchies exist that exclude individuals who are unfamiliar with the vocabulary and the jargon used by individuals positioned at the top [34]. Taking part in strategic decision making can be daunting [35] and unsurprisingly individuals unaccustomed to the language used will remain unable to influence decisions being made. Ironically, if service users become visible and confident enough to advocate for themselves, they tend to lose their status of being ‘representative’ of the user group. Technical and expert knowledge held by professionals is often weighted more highly than the practical and personal knowledge that members of the public may possess [36,37] and there is potential that the views of high-power/high-interest stakeholders are prioritised whilst the views of low-power/low-interest stakeholders may be ignored if time and resources run out [22,38]. Rose et al. accentuate the dilemma this creates by stating “Activists cannot speak on behalf of ordinary users and ordinary users cannot speak for themselves. In such a discourse, no service user can have a voice” [39]. A notion still exists that the technical and expert knowledge held by professionals is frequently held with higher regard than the personal experiences that lay stakeholders possess [30], hence their contributions may be unconsciously or even consciously downgraded [14,40].

Types of stakeholder involvement can vary significantly from a one-off consultation to engaging in continual exchanges of ideas and negotiation regarding the planning, design, and management of services [41]. Several studies report inconsistencies between the stated levels of stakeholder involvement that occur and the actual extent of involvement taking place in the multiple stages of the decision-making process [29,42,43,44]. Regarding alcohol services specifically, the 2018 community engagement in local alcohol decision-making (CELAD) study found some evidence of community engagement in local alcohol decision making; however, there was a lack of clear understanding regarding decision-making processes available to stakeholders and a lack of clarity regarding what input counts [45].

In the context of the study reported in this paper, the term stakeholder involvement was used to denote participation or engagement at any level in the design, development, and delivery of alcohol treatment services. This paper uses the example of alcohol services to explore the experiences and perceptions of stakeholder involvement in the planning, design and delivery of publicly funded services offering treatment and care to adults in need of support for alcohol-related problems.

## 2. Materials and Methods

### 2.1. Design

The research used a case study design within an NHS organisation within the North East of England, the NHS organisation encompassed three different local authority areas with a combined adult population (persons 16 years+) of 519,364 people [46]. This design was adopted to explore stakeholder knowledge and awareness of decision-making process and their experiences of being involved in alcohol service decision making within the boundaries of a specific environment or organisation [47,48] in this case three local authority areas in the North East Region of England. 

### 2.2. Qualitative Interviews and Focus Groups

Key stakeholder groups regarding alcohol decision making included alcohol commissioners; within the local authority, commissioners are responsible for meeting the treatment and care needs of their population through commissioning high-quality drug and alcohol services. Alcohol services deliver specialist support and treatment to individuals misusing alcohol. General practitioners, as a group of professionals, who may act as a first point of contact and referral source into specialist alcohol treatment and service users who have accessed support regarding their alcohol use. Qualitative semi-structured 1:1 interviews and focus group interviews were used to gain multiple perspectives from participants on whether stakeholders felt they had opportunities to be involved in decisions being made about alcohol treatment service provision. Semi-structured interviews were used with professional stakeholders, they provided enough flexibility to engage participants within busy work schedules and enable a wealth of information to be sought from participants. Focus groups were used with alcohol service users as they helped to stimulate discussions between group members and facilitate a range of responses in a relatively short space of time [49]. Furthermore, it was felt that the group environment would provide service users with ‘reassurance’ and enabled data to be sought without participants feeling pressurised to answer every question [50]. Barbour (2007: 26) suggest that conducting a group discussion allows “collective sense to be made, meanings negotiated, and identities elaborated through the process of social interaction with people”. It was also hoped that the focus group’s ability to generate a reflective participation approach would lessen any potential power differentials between the researcher and the participants due to promoting a ‘safety in numbers’ ethos, whilst also having the advantage of the group dynamic stimulating discussion [50]. Semi-structured topic guides were used in interviews and focus groups [51] to explore stakeholders’ awareness of opportunities that were available for them to be involved in decision-making processes, which mechanisms of involvement stakeholders had taken part in, the perceived barriers and facilitators of involvement mechanisms used and how much influence stakeholders felt they had regarding decisions made.

### 2.3. Sampling

Purposive sampling was employed with professional participants (alcohol commissioners and professionals within adult drug and alcohol services within the three local authority sites). The NHS organisation funding this study provided a list of all members of the alcohol commissioning team and all available adult drug and alcohol services within the case study sites, and therefore the population groups were pre-defined. Professional participants were recruited to ensure maximum diversity across the three local authority case study sites. Snowball sampling was used to recruit general practitioners (GPs). A single name of a GP with special interest in alcohol was provided by the NHS organisation, this GP then identified further colleagues with an interest in alcohol who may be willing to participate in this study.

Finally, typical case sampling [52] was used to recruit alcohol service users. The typical case within this study was defined as an individual in receipt of support via a specialist drug and alcohol service regarding their alcohol intake. Service users were invited to participate regardless of the level of support (prevention, cure, or treatment) they accessed [22]. Additional inclusion criterion applied to service users was to ensure that they were: 18 years old or over, alcohol was their primary substance misuse problem, and they could provide written informed consent. All professional participants were contacted directly by one researcher via email or telephone and the same researcher conducted the interviews. Service user participants were approached in person (face to face) by the service user involvement officer working across the case study sites. The focus group discussions were co-facilitated by the researcher and the service user involvement officer. Recruitment of all groups into the research continued until no new themes were arising from the data and data saturation had been reached [53].

### 2.4. Consent Process and Ethics

Once participants provided verbal consent to take part in the research, an appropriate time, date, and location were agreed to conduct the interview. Focus group discussions took place within active alcohol services, a service user involvement officer helped to recruit participants. Service users received a £10 voucher as remuneration for their time. Prior to every interview, a single researcher ensured that participants had read and understood the participant information leaflet, reiterated that participation was voluntary and provided an opportunity for any questions to be asked and answered. Ethical approval was gained from Sunderland Research Ethics Committee, currently re-named as North East Tyne and Wear South Research Ethics Committee (09/H0904/65).

### 2.5. Study Participants

The findings reported here are derived from an analysis of data obtained from 58 stakeholders, as shown in Table 1.

### 2.6. Data Analysis

All interviews and focus groups were audio recorded, transcribed verbatim and subject to iterative, in-depth, thematic analysis [54]. Thematic analysis was chosen as it allowed themes to be identified and explored as they arose during the analysis of the data. Initially, memos were used to capture any emerging ideas or questions of interest to the researcher. A constant comparison method of analysis [55] was adopted and ‘focused coding’ was used to identify key themes and concepts emerging from the data, which was compared within and across data sets. Data extracts were used to highlight similarities and differences across the transcripts and either challenge or evidence the emerging concepts. Qualitative software (NVIVO) aided the organisation of thematic codes. The main themes and findings were discussed and reflected on with the co-authors, and therefore the themes were constantly reassessed as new data emerged. The team involved in data analysis had expertise in public health, commissioning, and substance misuse research. The results of the analysis are presented below using verbatim quotes to illustrate the findings.

## 3. Results

Within the interviews and focus groups, all participants described opportunities for stakeholder involvement in terms of the mechanisms and approaches they encountered. Examples of stakeholder involvement varied from prescriptive, strategically imposed mechanisms such as a generic annual satisfaction questionnaire, to innovative opportunities for open-ended stakeholder involvement to occur. The analysis and discussion of participants’ reflections are explored in further depth below.

### 3.1. Prescriptive Methods of Involvement

Prescriptive methods of stakeholder involvement were expressed by participants as routine and systematic data collection, often used as part of a larger organisational review and/or as part of a performance management process. One such example described was a generic form completed at a scheduled time such as a specific point in a treatment process, i.e., an individual’s entry into or exit from an episode of care. Prescribed methods of stakeholder involvement were predominantly described as mechanisms that were imposed externally and included annual satisfaction surveys and questionnaires, and strategic groups and meetings.

#### 3.1.1. Annual Satisfaction Surveys and Questionnaires

Alcohol service managers presented examples of methods of stakeholder involvement that they carried out to satisfy external requirements. Such mechanisms consisted of completing documents such as a generic service evaluation forms or annual satisfaction surveys with pre-determined questions that were regulated at a national level and were issued to all alcohol service providers for completion. These standardised documents aimed to collate information for statistical purposes rather than to assemble information that reflected an individual’s journey and the associated experiences of treatment provision. Each alcohol service gathered information to help construct a national picture of the service users’ opinion regarding the alcohol interventions they had received. For some service managers, the purpose of using these standardised involvement mechanisms was to performance monitor their organisation, as Kat implies: “It’s done as part of the larger organisation and they will do a six-monthly service users surveys which is fed back up the chain if you like, so you know, so that they know we’re doing our jobs properly here”. (Kat, Service Provider).

This mechanism of involvement was descried as serving a purpose to feed information along the chain of command to commissioners regarding an individual’s ‘satisfaction’ regarding alcohol services. These mechanisms were portrayed as having limited scope for stakeholders to influence the design or delivery of services. Additionally, these mechanisms only permitted a snapshot of feedback to be acquired and they promote a one-way flow of information. It was not clear whether the information collected was made available or fed back to providers and service users and consequently the impact of this form of stakeholder involvement was uncertain.

Regardless of the unquantifiable scope of questionnaires and surveys, commissioning participants described them as suitable mechanisms to collect stakeholder views regarding service provision as emphasised by Laura: “We have a really good service user network in terms of we do a lot of questionnaires and I know people get sick of them but it’s about constant improvement and looking for different ways coz if it’s not working, they are the people that are gonna know about it”. (Laura, Commissioner).

In contrast, service users questioned whether surveys and questionnaires as methods of involvement were ‘fit for purpose’. Most of the service users identified that they had completed a questionnaire and/or survey whilst accessing alcohol treatment services. In contrast to the quote by Laura above, all service user participants felt that questionnaires were the wrong mechanism to collect quality and insightful views regarding services. The structure of these mechanisms of involvement were deemed to be too structured and they did not provide adequate opportunities for service users to express themselves. One participant explained that: “A questionnaire can’t get across the feelings and experiences, you can tick boxes but really, it’s not gonna give any more than that”. (Oscar, Service user).

Moreover, service user participants perceived that questionnaires did not reflect a genuine interest on behalf of decision makers to fully understanding individuals, views: “The questionnaires, the questions that they ask ya (you) and the boxes you’ve gotta tick, you probably tick the boxes but it’s not your true feelings, ya cannat express yourself in a ticky box situation, its limited to what they want to know, not what ya think ya want them to know”. (Tony, Service user). Service user participants articulated that a crucial component of any stakeholder involvement experience, was to feel as though their opinions and lived experiences had been heard and could potentially impact on future decisions. Service users suggested that this was the main aspect currently missing from current prescriptive methods of stakeholder involvement and this led to frustration regarding the perceived limited scope of involvement mechanisms to capture contextual and personalised information. Participants suggested face to face interactions would be more appropriate: “You don’t see the people behind the names, you just look at a load of questionnaires filled in by a load of people and there is still stigma attached to that. Whereas if you come here and you see the people face to face and you see that they’re just human beings like everybody else, you know. I think it kind of; it adds a lot more sort of strength to it than just words on a bit of paper”. (Emily, Service User).

#### 3.1.2. Strategic Groups and Meetings

From the perspective of alcohol service providers, they explained that they felt that contract review meetings that took place monthly between themselves and commissioners were one of the main opportunities for them to liaise with decision makers and be involved as stakeholders. It was uncertain how this mechanism of involvement influenced the decision-making process. “Every three months they’ll have a site visit where they’ll literally come in and walk around and look for evidence of what we’ve said we produce”. (Donna, Service Provider). Service providers perceived that structured methods of consultation were used to predominantly monitor their performance. From this perspective, there was limited opportunities for discussion to occur regarding how a service may evolve.

General practitioners also discussed their involvement in strategic meetings. From their perspective they described feeling as though they had been ‘consulted’: “I feel I’ve been consulted appropriately…. What I can’t influence is their ultimate decisions say on financial matters; and that’s fair enough, that shouldn’t be my decision”. (George, General Practitioner).

From a different perspective, one GP felt that their voice had the potential to be excluded due to their dual role as both a clinician and a stakeholder; “My Clinical Lead voice has been excluded but then I mean I understood you know why there had to be this you know because I am a provider as well”. (Greg, General Practitioner).

Predominantly, methods of involvement were described from the professional’s perspective as occurring at the level of consultation, implicitly suggesting that it had limited impact on decision-making processes. However, genuinely inclusive methods of involvement were described in one location (service B) where service users explained that service managers held regular committee meetings that were open for services users to attend: “I would say, given the fact that this is, it’s like the meeting every 2 months with the powers that be (service managers) and that’s another level of getting things passed on to management or whatever, coz what it is basically is that they sit and listen to everybody”. (Sarah, Service User). These open meetings provided an environment where service users and individuals volunteering within the service could communicate their views regarding alcohol service design on a regular and ongoing basis.

### 3.2. Involvement Mechanisms with Potential for Innovation

Despite prescriptive involvement mechanisms being perceived to have limited levels of influence on overall design and delivery of alcohol services, some participants did describe opportunities for increasingly ‘authentic’ stakeholder involvement to occur. When discussing the emergent informal mechanisms of involvement, participants described a more spontaneous exchange of ideas taking place. Processes occurred that included direct and personalised consultation and user-led innovation that were responsive, and needs led rather than being imposed at pre-specified times.

#### 3.2.1. Direct Consultation

Service providers identified increasingly inclusive methods of stakeholder involvement at a local service provider level with Kat stating that: “I think it’s very important to listen to the client, you know they’ve got the answers so we’ve just introduced part of our new recruitment strategy is we have a client or couple of clients showing the candidates round giving you know the tour of the place and informing them how things run in here and then we sit at the end of the interviews with the clients and take feed-back from them and what they thought about the candidates”. (Kat, Service Provider).

This example highlights that service providers and service users jointly influenced decisions being made regarding professionals employed in alcohol services, this created a sense of empowerment within individual services.

Most services providers identified that stakeholder involvement took place within their individual organisations. At this local level, ‘informal’ involvement opportunities were described as being continuously available and experiences of service provision were gathered using multiple methods such as service user feedback forms, suggestion boxes, evaluation sheets and patient stories. Services portrayed a commitment to providing several methods for individuals to communicate their experiences; “there’s various ways and methods that people can communicate which will make them feel most comfortable”. (Donna, Service Provider). The availability of different methods of involvement indicated a deliberate attempt to offer service users a format that was appropriate for everyone. For example, service users were able to submit proposals or suggestions regarding service design or delivery anonymously. Additionally, if service users preferred to communicate face to face, weekly community meetings were scheduled with a service manager. However, the impact that these involvement mechanisms had on overall decision making was difficult to measure, as despite multiple consultation opportunities existing within a single agency, and participants discussing them positively, participants did not explicitly state how these views influenced the final decisions (or how they were utilised within decision-making processes).

Two alcohol agencies described a less organised approach to service user involvement. Within these two agencies, involvement appeared to take place in a more reactive and unplanned manner, responding to specific needs as they were identified: “It hasn’t happened very much other than the anecdotal stuff where a client might come in and say well that so and so is great or that’s crap you need to do something about that and then the organisation will respond you know in an anecdotal way a very ad hoc way”. (Seth, Service Provider). Although it could be perceived as negative that service user involvement did not happen in a continuous way, it also highlighted the level of flexibility that services have, to be able to instigate change in response to suggestions made regarding their service offer.

#### 3.2.2. User-Led Innovation

There were two notable examples where service user participants described their recovery journey and their progression from patient or client to service manager. In these examples, individuals who were current alcohol service managers drew on their own personal treatment journey through alcohol services—they had recognised areas of unmet needs and had therefore identified potential areas for alcohol service redesign. What these examples illustrate is that regardless of an individual’s status, it is possible to influence decision making regarding service design and delivery if opportunities are available and the necessary evidence is collated to identify a gap in the current service provision: “The services was actually born from a group of individuals in recovery from alcohol and drug addiction and they identified there was a gap in this style of treatment so they met on a voluntary basis over a two year period you know looking at a model in which they could work... so it’s actually come from the service level upwards”. (Rebecca, Service Provider). In one example, William described a prolonged period of self-funding a potential alternative treatment option, to demonstrate its need to commissioners: “Service B was registered as a charity in 2005, it started somewhat eighteen months before that as, a support group in my home, and for two and a half years I funded it totally out of my incapacity benefit, yeah, yeah. That’s how it started”. (William, Service Provider). The two examples above highlight instances where evidence was collated to demonstrate that new elements of service design were needed to respond to unmet need and that a different service offer constituted a legitimate addition to the alcohol treatment pathway.

## 4. Discussion

Analysis of the data from our qualitative case study of alcohol services in the North East of England shows a lack of consistency regarding the opportunities within the decision-making processes for stakeholder involvement to occur [56].

Prescribed mechanisms were described as occurring at pre-determined time points and were regulated at a national level, leaving little scope for localised nuances to be captured. Data were portrayed as being collated to help construct a national picture of what was happening across service providers, with limited scope for stakeholders to influence design or delivery of services. As described in the findings, service B was an exception to this and it was expressed that service B involved stakeholders in a way that felt genuine, so individuals felt as though their views were heard and actioned. Differences in opinions were present, as commissioners described survey and questionnaires as appropriate mechanisms to obtain service users opinions, whilst the service users themselves portrayed them as the wrong mechanisms, lacking the ability to collect quality and insightful views.

Stakeholders continued to illustrate emergent informal mechanisms of involvement within individual services, these methods were described as responsive, and needs led rather than imposed at pre-specified times. The potential for this level of stakeholder involvement to be influential was more prominent at a localised level where there was recognition that service users have ‘the answers’ and multiple opportunities to become involved were present such as feedback forms, suggestion boxes, evaluation sheets and patient stories. All the mechanisms were available to complete on a continuous ongoing basis, providing an open dialogue with the potential for changes to occur to alcohol service provision.

Stakeholders falling into the professional categories such as alcohol service providers and GP’s perceived involvement opportunities as a mechanism to monitor their performance and were predominantly depicted as ‘consultation’. Professional stakeholders also described a silo-based approach to involvement in decisions, where decisions were made in relation to individual organisations rather than to address system wide priorities; an issue that persists today [57,58].

The majority of stakeholders described involvement that was sporadic (sometimes happened and sometimes did not), subsequently, it was hard to ascertain within each scenario, the level of impact that stakeholder involvement had made to alcohol services [22]. Aside from two notable examples, our study found few instances of stakeholder involvement that unequivocally influenced the design and delivery of drug and alcohol services. The findings showed that commissioners controlled the budgets and final decision making. However, it could be argued that due to continuing austerity, severe financial pressures impact upon every part of the health and social care system [57]. This is especially true within alcohol and drug treatment services that have seen a £212.2 million disinvestment between 2013/2014 and 2018/2019 [59] which will result in stakeholder involvement not receiving as much time or investment as is necessary and enabling prevailing power structures to remain intact.

Pennington et al. (2018) acknowledge that despite intentions of meaningfully involving stakeholder in decision making, evidence regarding the impact of this involvement is still limited [5,35,60]. Even when stakeholders were given an opportunity to participate in decision-making processes, we found little consensus regarding the actual influence such involvement mechanisms had regarding service design and delivery [22,61,62]. Indeed, this reflects a wider gap in published evidence about what processes and mechanisms are most effective in ensuring multiple stakeholders’ voices are heard within the decision-making processes [61,63,64]. A study by Rushmer et al. [56] found that members of the public did not have a ‘direct voice’ regarding the implementation of public health interventions to reduce alcohol-related harm. However, it should also be acknowledged that stakeholder involvement in alcohol services may present additional challenges. Alcohol is a public health issue that impacts on a wide variety of individuals, inclusive of social drinkers, individuals who are experiencing alcohol-related harm themselves and family and friends of people experiencing alcohol-related harm [65]; however, many of these individuals would not consider themselves to be a ‘stakeholder’. For alcohol service users, their personal ‘lived experience’ is valuable and important [66]; however, service user participants within this study found it difficult to achieve sufficient distance from their own health challenges to be able to speak on behalf of other service users, both known and unknown, to help shape wider decisions regarding the necessary alcohol service provision [22]. Alcohol misuse can also be sensitive to discuss due to perceptions of vulnerability and potential negative judgement [67]. This raises legitimate concerns regarding their ability to contribute to wider health care concerns such as resource implications [30]. This concern is of course not limited to health care and there is a broader debate about the extent to which public involvement should be used to inform professional practices and policy. The same apprehensions are echoed regarding shaping criminal justice policy [68] and social care reform [69] due to individual stakeholders having differing priorities often based on their own narrow personal experiences.

Limited literature is available regarding stakeholder involvement in decision-making processes regarding alcohol service design and delivery. Our data reinforced previous findings in the CELAD study which found evidence of involvement at a local level but also concluded that most mechanisms were ‘top down’ and were managed by local authorities. Our data and previous findings imply a limited level of control regarding how stakeholders are involved in the decision-making process [35,70]. Providing an opportunity may not be sufficient to ensure an individual’s ‘expertise’ is seen as legitimate or influential [36,37] as reported elsewhere [30,31,71]

### 4.1. Strengths and Limitations

A strength of this study is that it collected data from multiple perspectives and allowed data to be compared and contrasted across participant groups. The focus of this study was on decision making regarding alcohol service provision which is an under researched area. The issues presented in this paper and the need to develop timely and appropriate mechanism for involvement to occur are significant due to challenges arising due to prolonged austerity and constrained resources for stakeholder involvement to occur [35]. Limitations were also present; despite a blanket invitation to alcohol service users regardless of the type of support they accessed (brief intervention, advice and support, therapeutic intervention), the participants who attended the focus groups were currently accessing or had accessed services offering social and psychological interventions, medical detoxes and/or residential rehabilitation. The limited recruitment of service users from the entire spectrum of alcohol treatment provision could mean that the findings are not representative of all stakeholders who could access alcohol support. Additionally, data collection only took place in the North East of England and therefore a common criticism is that it provides a poor representation of the wider population [47]. However, whilst recognising this criticism, it should be acknowledged that generalisation was not the aim of this study. The primary purpose of this paper was to capture the complexity of a situation occurring within and across the case study sites, to enable local decision makers to reflect on their practice [22,47].

### 4.2. Implications for Policy, Practice, and Future Research

We found that most alcohol stakeholders do not perceive themselves as influencing decisions relating to service design and delivery as fully or consistently as policy proposes that they should [22]. Clear guidance of potential opportunities for stakeholder involvement needs to be established, and these need to have more dimensionality than stakeholder involvement policies currently afford [72,73,74]. More needs to be done to support and encourage engagement of wider stakeholders in alcohol decision making inclusive of commissioning processes and service design. To enable this to happen, decision makers need to think about the most appropriate methods of communicating information regarding stakeholder involvement opportunities. There are established communities for individuals in alcohol recovery, and people within these networks could contribute usefully to the alcohol decision-making process.

More recently, the introduction of integrated care systems in England and devolution across the UK has aimed to increase the focus on planning, collaboration and working across a whole system, the way decision making occurs is continually evolving [75]. Arguably, this move to promote collaboration more explicitly should be supportive of improved stakeholder engagement. To achieve meaningful collaborative working, ideally a co-production approach should be taken, in which decision makers, members of the public and professionals work together to share power and responsibility for the decisions being made. Decision makers need to avoid tokenism. This can be achieved by explicitly sharing knowledge and information about how to engage with the alcohol decision-making process and being transparent regarding which opinions and interests are reflected within final decisions that are made.

Many participants expressed their disappointment and frustration at the regular use of overly prescriptive methods of stakeholder involvement preferring methods that enabled mechanisms to be ongoing and responsive to the environment and context that the involvement was occurring within. There needs to be increased availability of ‘bottom up’ engagement opportunities for stakeholders to participate in, inclusive of alcohol service users at different stages of recovery, significant others (related to the alcohol user), members of the local community and key professionals. Policy documents need to be developed that consider promoting more appropriate and sustainable mechanisms to involve stakeholders within all stages of the decision-making process. Adopting co-production principles inclusive of sharing of power, ensuring all perspectives are represented, respecting and valuing everyone’s knowledge, reciprocity, and building and maintaining relationships [76] could enable this to happen.

Further research using realist evaluation methods to explore how integrated care systems have influenced how care is planned and delivered within complex health care settings such as alcohol and substance misuse would be useful. Once more evidence is available and has been critically analysed, clear ‘best practice’ guidance needs to be developed to encourage stakeholder involvement to become standardised within specialist public health services such as alcohol treatment services.

## 5. Conclusions

All participants interviewed as part of this study were stakeholders. However, for some participant’s, stakeholder involvement remains rhetorical, and the understanding of what stakeholder involvement comprised of varied among participants, with some definitions implying that involvement only took place at ‘consultation’ levels. Professional participants (including commissioners and alcohol service providers) inferred that stakeholder involvement was often regarded as a ‘means to an end’ rather than a continuous process, and this was highlighted by the heavy reliance on tick box questionnaires—the insinuation being that ‘any involvement was good involvement’.

Stakeholder participants identified multiple opportunities for genuine and informative involvement to take place; nevertheless, it was hard to establish whether involvement mechanisms influenced planning and decision making that took place. It is essential within public health that a deeper understanding is generated of who stakeholders are within this complex and fluid environment and to develop a clearer understanding of the different roles that stakeholders can play within the entire decision-making process in order to maximise its utility.

## Figures and Tables

**Table 1 ijerph-19-02148-t001:** Study participants.

Participant Group	Demographics	Local Authority 1	Local Authority 2	Local Authority 3	Total
Alcohol service users	Male (n = 18); Female (n = 13)	11	12	8	31
Service providers	Male (n = 4); Female (n = 7)	2	2	2	10 (4 providers offered services across all 3 areas- the strategic manager was interviewed)
Alcohol commissioners	Male (n = 6); Female (n = 4)	4	3	2	11 (2 commissioners worked strategically across all 3 areas)
General practitioners	Male (n = 5); Female (n = 1)	2	2	2	6
					58

## Data Availability

The data sets used and/or analysed during the current study are available from the corresponding author on reasonable request.

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
