# Peer review of "A Qualitative Exploration of Stakeholder Involvement in Decision-Making for Alcohol Treatment and Prevention Services"

_ijerph, 2022, doi:10.3390/ijerph19042148_

Round 1

Reviewer 1 Report

Overall this is a very well-written and interesting manuscript among a varied sample of relevant stakeholders, which I anticipate will be of great interest to the journal's readership and beyond. Subject to the minor revisions/additions described below, I support the publication of this manuscript within this journal in its present form.

Intro – Overall the introduction of this manuscript is very well written and provides a comprehensive account of literature in this field. That said, to ensure the literature review is as contemporary and complete as possible and includes broader links to literature from other disciplines, the authors would benefit from also referring to the recent intervention research/reviews from the discipline of psychology where pro-social attitudinal and behavior change interventions also prioritize PPI and stakeholder involvement. As such, at the end of the sentence on line 45-48,  ‘within English health care policy for several decades (2-4)’ the authors should add in something along the lines of ‘and more recently within disciplines such as psychology, aiming to affect behavior and attitude change’ with reference to relevant research e.g. Hudspith et al (2021) DOI: 10.1177/15248380211050575 [attitudinal change interventions] and Boduszek et al (2019) DOI:10.1016/j.chb.2018.12.028 [behaviour/cognition change intervention].

Methods – The authors explain exceptionally well the study design and sampling procedures. Likewise, the method of analysis was clearly articulated allowing replication of the study in the future and sufficient explanation to warrant publication in the current manuscript. That said, there are two minor additions/clarifications/expansions needed to sufficiently justify and explain the approach taken.

  • Line 124-126 – include a citation to make clear why this approach adopted was likely/considered to reduce power differentials rather than exacerbate them. Intuitively it appears as though the opposite could equally be true.
  • Line 166 onwards – the authors would benefit from providing a very brief justification for use of this method of analysis given that alternative approaches could also prove useful here e.g. conversation analysis etc. As such, briefly outline why TA was deemed the most suitable approach given the aims of the research.

Results – detailed and insightful result section reported in line with convention where quotes are embedded very well within broader sub-sections as evidence of each particular theme.

Discussion – The authors provide an in-depth and comprehensive step-by-step account of the current findings in the context of previous theorizing and study findings – an excellent discussion overall. That said, whilst not exclusively related to public health, there is, of course, a broader debate about the extent to which public involvement, opinions, and decisions should be used to inform professional practice and policy given that misconceptions, stereotypes, and narrow experiences may influence PPI decision making in a way that it is hoped professionals would not. Criminologist Olivia Smith (Smith et al (2021) DOI: 10.1332/239868020X16057277095797) has recently considered this is in the context of whether public rhetoric should ever be used to shape criminal justice policy and it would make therefore for an interesting addition here if the authors could include a brief comment or acknowledgment of this argument/debate more broadly, to ensure the article has wider applications beyond the discipline of public health (line 419 or perhaps line 450 onwards).

Author Response

Thank you for your helpful comments, please see attached responses to your comments.

Reviewer 2 Report

The submitted manuscript is a qualitative exploration of stakeholder involvement in decision making process for alcohol treatment and prevention services in the UK. Overall, this article tackles an important and interesting subject as stakeholder and user involvement in policy and decision making is often outlined in many guidelines, however rarely takes on the optimal form in reality. As the authors themselves note very clearly. In terms of the qualitative analysis, this is a good article, there are some issues that warrant attention:

  1. First paragraph of the introduction there seems to be a mishap with the references as the first reference that appears is numbered 7 and seems to be in a different citation form compared to the rest of the text.
  2. I suggest you take a look at the sentence where you define "stakeholder" as it makes little sense as it is written. 
  3. Design section could use more data on the hierarchies at play. You are not writing to an audience that is familiar with inner functioning of UK services and the paper should reduce the UK-centrism. What is the relation between a "commissioner" and a "service provider"? Readers cannot place your data or understand it if these relations are not explained. 
  4. Table 1 please provide footnotes for abbreviations. What is PCT? In the demographics please add mean age of the participants. 
  5. While I do understand the need to present data verbatim, the authors may consider to edit some of the colloquialisms used in citations as they may be problematic for some readers. 
  6. The discussion could benefit from some specific data and suggestions in terms of alcohol treatment and prevention services. Seems as an afterthought (may explain the mishap with the references in the introduction). 

Author Response

Thank you for your comments. Please see attached document to see our responses.

Reviewer 3 Report

The authors report on a qualitative study of stakeholder involvement in, and influence upon decisions about alcohol prevention and treatment services in the UK. The study addresses a critical emerging focus in public health in the UK, as reflected by the National Health Service's (NHS) long-term plan to increase stakeholder involvement in decision-making about health and care service provision. The aim of the study was a deep exploration of stakeholder involvement in the specific case of public health alcohol services in three local authority areas in the North East of England.

The methods section describes using purposive, snowball, and typical case sampling to identify study participants from different stakeholder groups, including alcohol commissioners and alcohol service providers, general practitioners, and alcohol service users, respectively. Data were collected from 58 participants through interviews and focus groups, then transcribed and analyzed thematically. It is not clear which of the authors conducted the interviews and focus groups, but the statement "All professional participants were contacted directly by the researcher (emphasis mine) via email or telephone (line 149-150)" implies a single interviewer or group facilitator. Similarly, it appears (but is not clear) that a single author conducted thematic analysis, developing codes and themes, then shared it with co-authors. The theoretical or conceptual framework underlying the methodology for analysis and consensus-building was not adequately described, making the process less transparent and reproducible.

The authors provide a good background and description of stakeholder involvement and its benefits and challenges in the introduction. However, the introduction did not make a convincing argument for the significance of the study. It is not made clear how the present study will significantly add to the existing knowledge about the complex nature of stakeholder involvement or whether it may identify unique concerns about such involvement (or lack thereof) within alcohol services compared to other types of public health services. Nevertheless, the results are reported in a clear and organized fashion and appear to support claims made in the introduction about the difficulties inherent in stakeholder involvement.

In the discussion, the authors address the possible "additional challenges" of involving stakeholder alcohol services, especially service users. As stated above, if emphasized in the introduction, this could make a stronger case for the significance of the study and help shape its overall focus. In addition, some parts of the discussion seem more appropriate for the introduction, for example, the paragraph inclusive of lines 401-424. The implications section beginning at line 444 had good suggestions, but I was hoping to see more specifics, especially some arising from the findings in the present study.

Author Response

(The authors gave the same response as above.)

Round 2

Reviewer 2 Report

Thank you, my comments have been sufficiently addressed. I have no further comments.